# The interfacial structure of water droplets in a hydrophobic liquid

Nikolay Smolentsev[1], Wilbert J. Smit[2], Huib J. Bakker[2] & Sylvie Roke[1]

Nanoscopic and microscopic water droplets and ice crystals embedded in liquid hydrophobic surroundings are key components of aerosols, rocks, oil fields and the human body. The chemical properties of such droplets critically depend on the interfacial structure of the water droplet. Here we report the surface structure of 200 nm-sized water droplets in mixtures of hydrophobic oils and surfactants as obtained from vibrational sum frequency scattering measurements. The interface of a water droplet shows significantly stronger hydrogen bonds than the air/water or hexane/water interface and previously reported planar liquid hydrophobic/water interfaces at room temperature. The observed spectral difference is similar to that of a planar air/water surface at a temperature that is ~50 K lower. Supercooling the droplets to 263 K does not change the surface structure. Below the homogeneous ice nucleation temperature, a single vibrational mode is present with a similar mean hydrogen-bond strength as for a planar ice/air interface.

[1] Laboratory for Fundamental BioPhotonics (LBP), Institute of Bioengineering (IBI), and Institute of Materials Science (IMX), School of Engineering (STI), and Lausanne Centre for Ultrafast Science (LACUS), École Polytechnique Fédérale de Lausanne (EPFL), CH-1015 Lausanne, Switzerland. [2] AMOLF, Science Park 104, 1098 XG Amsterdam, The Netherlands. Correspondence and requests for materials should be addressed to S.R. (email: sylvie.roke@epfl.ch).

Aqueous hydrophobic interfaces play a defining role on nano- and microscopic length scales. Pockets and droplets of water in a hydrophobic environment are omnipresent in the atmosphere (as ice particles and cloud droplets[1]), the earth[2,3] (in oil fields and inside internal pores of many geological materials), and in chemical[4,5] and biological processes[6–8] (as vehicles for medicine delivery[9]). For all of these processes, the nanoscopic water droplet/hydrophobic interface plays a critical role in determining the fate of the system. The reactivity of confined water is crucial for atmospheric science[1], earth science[3] and biology[10]. The molecular surface properties of water droplets are commonly inferred from data obtained from the macroscopic air/water interface[11–16] or from aqueous solutions of solvated hydrophobes[17], systems that are very different in both their size, chemical composition and temperature dependence from actual droplet surfaces. As such, the surface structure and reactivity of water droplets and confined water remains elusive, despite its relevance.

Here we present temperature-controlled sum frequency scattering (SFS) experiments on nanoscopic water droplets in a liquid hydrophobic environment. We find that compared to planar water/air or water/hexane interfaces the spectrum reveals a greater order corresponding to that of a planar water surface at a ~50 or 40 K lower temperature. This increase in order is explained by the formation of an extended hydrophobic network, which does not exist on planar interfaces and in solution. In addition, on supercooling the droplets, the surface spectrum does not change shape. Cooling the water droplets below the homogeneous ice nucleation temperature results in a spectrum with a single symmetric peak that is similar to the one found for the basal ice/air interface.

## Results

**Room temperature droplet experiments.** Nanoscopic (100 nm radius) water droplets are prepared via sonication in $d_{34}$-hexadecane (Fig. 1) or in a 1:1 mixture of decane and cyclohexane (Fig. 2), both with 5 mM of a hydrophobic surfactant molecule that partially covers the interface (Span80, Fig. 1a). The water–hexadecane partitioning coefficient for these hydrophobic liquids is $5.5 \times 10^{-4}$ (Span80 (ref. 18)), $> 1.6 \times 10^{-6}$ (decane) and $1.6 \times 10^{-4}$ (cyclohexane)[19], providing a liquid hydrophobic/water interface in all systems. The interfacial structure of the water droplets and ice nanocrystals is measured with vibrational SFS[20,21], a method that combines interface-specific sum frequency generation (SFG)[11,12,14,15,22,23] with light scattering[24] (illustrated in Fig. 1b). The scattered vibrational spectra report on the average orientational order and H-bond network structure of water molecules in the first few monolayers of water at the interfacial region[21].

Figure 1c shows the scattered SF spectrum of $D_2O$ nanodroplets in oil (296 K) and reflection SFG spectra of a planar liquid $D_2O$/air and a planar $D_2O$/hexane interface. The spectra were normalized taking into account linear absorption effects as well as discontinuities in the normal component of the electromagnetic fields as they cross the droplet interface (see Methods section). The SFG spectra of the two planar interfaces of Fig. 1c have three main features: A peak at $2,370/2,400 \, cm^{-1}$, a peak at $2,500 \, cm^{-1}$ and a peak at $2,745/2,725 \, cm^{-1}$. The frequency of the OD stretching vibrations is correlated to the strength of the interfacial hydrogen (H)-bond interaction: the OD frequency decreases with increasing H-bond strength. Hence, the feature at $2,500 \, cm^{-1}$ reports on water molecules that are more weakly H-bonded than the ones that attribute to the feature at $2,370/2,400 \, cm^{-1}$. The peak at $2,745/2,725 \, cm^{-1}$ originates from interfacial OD bonds that are not H-bonded[25]. The 296 K droplet/hydrophobic interface spectrum contains the same 2,370 and $2,500 \, cm^{-1}$ features (indicated by the dotted lines), but the higher frequency components are significantly less intense than they are in the planar interface spectra. This indicates that there are relatively more strongly H-bonded water molecules at the water droplet interface than at the air/water and hexane/water interface recorded at the same temperature. The SFG spectrum of the Span80/water interface is similar to that of $D_2O$/hexane and $D_2O$/air and shown in the Supplementary Fig. 1.

The peak ratio of the 2,370 and $2,500 \, cm^{-1}$ modes is temperature dependent. This temperature-dependent ratio is plotted in Fig. 1d for the air/water and hexane/water interfaces. Lowering the temperature increases[26] the ratio, indicating that the population of stronger H-bonds increases over that of weaker H-bonds. The peak ratio obtained for the water droplets is also indicated (black marker). Extrapolating the ratios found for the air/water and hexane/water spectra (blue and red lines) the same ratio as for the water droplets would be found at a hypothetical planar water/air or water/hexane interface of ~245/260 K (that is, ~50/40 K colder). Thus, the water molecules of the water droplet/hydrophobic interface appear to be much more structured than the water molecules at a planar water/air or water/hexane interface. Note that this conclusion was arrived at by comparing amplitudes at particular frequencies. The spectrum of water is known to be highly complex, involving strong inhomogeneous broadening[27,28], excitonic coupling effects[29] and a Fermi resonance with the overtone of the bending vibration[30]. As such, aside from the feature that high-frequency modes generally refer to more weakly H-bonded water molecules than low-frequency modes, there are many ways to interpret the spectral information[15,16,25,26,28,31–35]. Our analysis addresses the temperature dependence of the ratio of weakly and strongly H-bonded water molecules, and is as such quite model independent.

Very similar SFS spectra are obtained for other combinations of hydrophobic oils (mixtures of decane and cyclohexane, fluorinated oil) and surfactants (Tween80, tri-block PEG 900, Supplementary Fig. 2), which indicates that the observed red-shift of the spectrum is a common phenomenon for water droplets in various hydrophobic liquids. The absence of unbonded OD groups in the droplet spectrum can be partially explained by the presence of OH groups on the Span80 molecule that can form H-bonds with interfacial water molecules that would otherwise be unbonded (see Supplementary Note 1, Supplementary Fig. 3 and Supplementary Table 1). As a result, the peak at $2,745 \, cm^{-1}$, corresponding to OD groups that are not H-bonded, vanishes. From the detection limit of the SFS system[36], it follows that the imbalance between OD groups that point in or out of the water droplet surface is less than one free OD group per $27 \, nm^2$. The difference with the droplet spectrum probably arises from a difference in structure, which we will discuss in more detail further on. Note that impurities are not expected to be a problem here[36] as the sample volume is small (~50 μl), and the surface to volume ratio is large (~$10^5 \, cm^{-1}$; which is at least three orders of magnitude larger than in standard planar reflection SFG measurements).

**Supercooling the droplets.** Next, we repeat the experiment at reduced temperatures, supercooling the droplets (Fig. 2a) and ultimately freezing them by cooling below the homogeneous nucleation point of $D_2O$ (237 K (ref. 37), Fig. 2b). Figure 2a shows SFS spectra recorded at different temperatures (296 K, red, 273 K, green and 263 K, blue) using a mixture of decane and cyclohexane as the main phase. The droplet spectrum from Fig. 1c is also shown (black line). It can be seen that despite the different

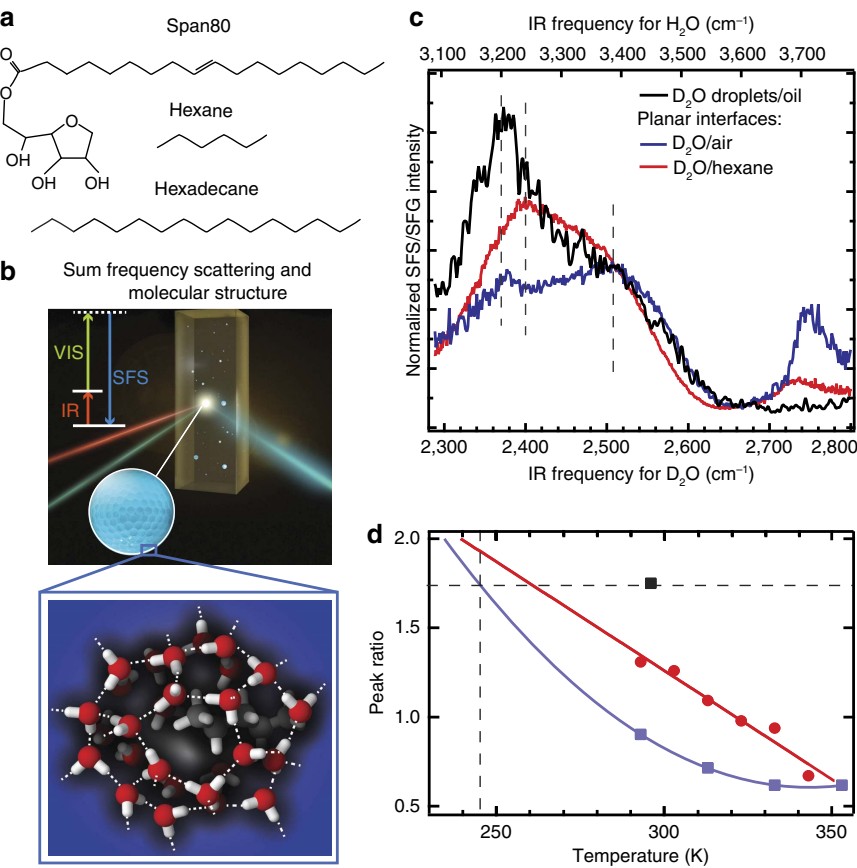

**Figure 1 | Surface structure of water droplets.** (**a**) Chemical structure of hexadecane, hexane and Span80. (**b**) Illustration of a vibrational SFS experiment. A femtosecond infrared (IR) and a picosecond visible (VIS) laser beam overlap in space and time in the sample and generate scattered sum frequency light from the water droplets. The inset illustrates the surface hydrogen-bond structure of a water droplet embedded in a hydrophobic liquid. In this figure, we sketch a possible molecular arrangement of the water molecules at the surface of a droplet surrounding an alkane tail that protrudes into the water phase (looking from the inside of the droplet). Please note that the surfactant is not present in the illustration. (**c**) SFS spectra of $D_2O$ droplets in $d_{34}$-hexadecane with 5 mM Span80 (black) and reflection SFG spectra of the planar $D_2O$/air interface (blue) and planar $D_2O$/hexane interface (red). The SFS/SFG spectra are collected with horizontally (P) polarized infrared and vertically polarized (S) VIS and SF beams. The normalization procedure takes into account the infrared pulse shape in the sample as well as discontinuities of the electromagnetic fields at the interface (see Methods section). The top axis shows the corresponding frequencies axis for $H_2O$ ($\times 1.35$[15]). (**d**) The ratio between the low- and high-frequency bands of the SFG spectrum of the planar water/air and water/hexane interface[26] (blue and red markers) and an extrapolation with a quadratic polynomial fit to a lower temperature range (blue and red lines). The ratio for the room temperature water nanodroplet spectrum is shown as a black marker.

hydrophobic phase the water spectrum does not show a significant difference. In addition, reducing the liquid temperature below 273 K does not result in any spectral changes. Figure 2b shows the SF spectrum of $D_2O$ nanocrystals (233 K) and the ice/air interface for different temperatures (170, 200 and 230 K) in the frequency range around the strongly H-bonded mode at $\sim 2,350$ (3,170) cm$^{-1}$. The resonance in the SF spectrum of the ice nanocrystals has a similar frequency as is observed for the planar ice surface of the same temperature. This indicates that the difference in surface structure observed for the liquid interface is not present in the solid phase. The absence of the effect for the frozen droplets can be understood from the fact that when the droplets freeze, the mutual water–water interactions become relatively more important for the resulting water SFG spectrum (and the detected H-bond strength) than the interactions with the hydrophobic oil molecules. For crystalline planar ice, the observed SF ice/air spectrum originates in part from OH groups connecting the different layers in the ice crystal[38]. These deeper layers of water molecules are much less influenced by the hydrophobic groups at the interface. Hence, for planar ice and frozen water droplets of the same temperature the H-bonds of the probed OH groups are of similar strength.

## Discussion

Thermodynamic[39,40] and Raman multivariate curve resolution (MCR) measurements on solvated alkanols[17] show water molecules hydrating a hydrophobic solute are more tetrahedrally ordered and thus more constrained in configurational space than water molecules in bulk liquid water (as sketched in the inset of Fig. 1b). These solutes are very small and fulfil the criterion of Chandler[4] that states that hydrophobic solutes/surfaces lead to an enhancement of the tetrahedral ordering of water if the radius of the solute/surface is <1 nm (ref. 4). In this perspective, the present observations of an enhanced ordering of water molecules at the surface of water nanodroplets embedded in oil is perhaps surprising, as the droplet radius (100 nm) is much larger than 1 nm. However, we can consider the oil surrounding the water droplet surface as an extended curved array of hydrophobic perturbations, that is, hydrophobic groups protruding into the surface of the water droplet. The water molecules near the hydrophobic protrusions will fold their H-bond network around these (<1 nm) protrusions. We speculate that a high density of hydrophobic protrusions in combination with a modest overall curvature of the water surface (corresponding to a diameter of a few hundred

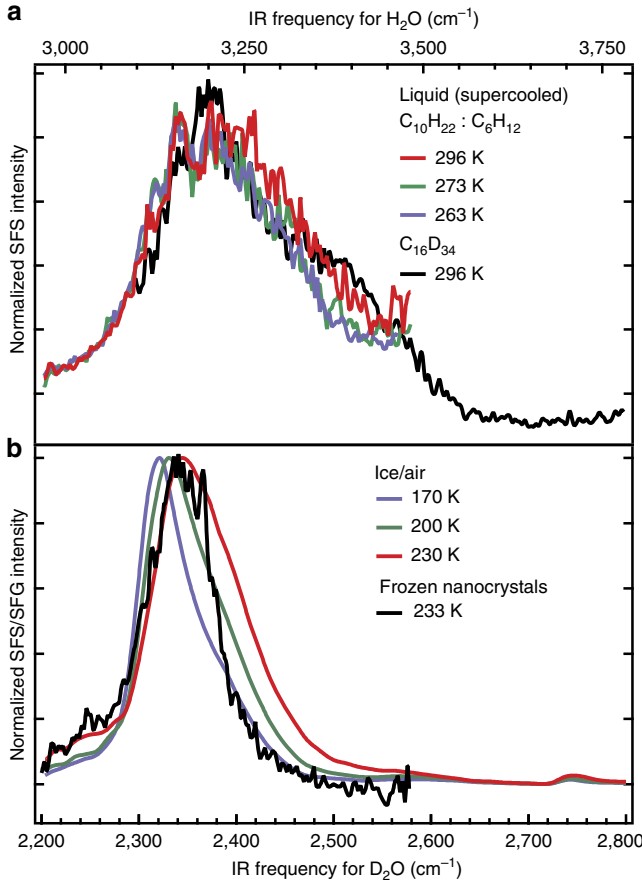

**Figure 2 | Supercooled water droplets and ice nanocrystals. (a)** SFS spectra of $D_2O$ droplets at room temperature (296 K, in a mixture of decane and cyclohexane (red) and in $d_{34}$-hexadecane (black)) and under supercooled conditions (green, 273 K and blue, 263 K, both in a mixture of decane and cyclohexane). **(b)** SFS spectra of $D_2O$ droplets at 233 K, below the homogeneous ice nucleation temperature (black) and SFG spectra of the basal $H_2O$/air interface at 170 K (blue), 200 K (green) and 230 K (red). All droplets samples contain 5 mM Span80. Spectral data from $H_2O$ and $D_2O$ are interchangeable. Frequency conversion between $H_2O$ and $D_2O$ was obtained by multiplying the $D_2O$ frequencies by 1.35[15]. The SFG spectra are collected with horizontally (P) polarized infrared and vertically (S) polarized VIS and SF beams.

nanometres), leads to a strong structuring of the water hydrogen-bond network, resulting in an extended highly structured corrugated droplet surface. For smaller droplets (for example, within micelles/solutes), the surface is too strongly curved to display such a favourable interplay between short-range and long-range interactions. For larger droplets, and flat surfaces, the curvature is too weak to get such a favourable interplay, leading to local disruptions of the H-bond network and an enhanced inhomogeneity. This picture of the enhanced ordering of curved water surfaces will require support from high-quality molecular dynamics simulations.

In summary, vibrational SFS spectra from 100 nm radius water droplets embedded in a hydrophobic liquid at different temperatures show that the H-bond network of the interfacial water possesses a much enhanced (in-plane) tetrahedral structure compared to a planar air/water interface. This greater order manifests itself as a red-shifted spectrum that has a 100% increased peak ratio of the 2,370 and 2,500 $cm^{-1}$ modes. This increase is much larger than observed in previous studies of planar liquid hydrophobic/water interfaces. The observed

structure of the water droplet interface corresponds to that of an air/water interface that is ~50 K colder. This increase in order is explained by the formation of an extended network of hydrophobic protrusions into the water droplet surface, which does not exist on planar interfaces and in solution. On supercooling the droplets, the surface spectrum does not change shape. Cooling the sample below the homogeneous ice nucleation temperature results in a spectrum with a single symmetric peak at a comparable frequency as that of a planar basal ice/air interface of the same temperature.

The presented experiments demonstrate the possibility of quantifying the interfacial structure of water droplets and illustrate the effect of nanoscopic hydrophobic surfaces on the H-bond structure of water. With a reduction of the effective 'surface temperature' we expect that the reactivity of water droplets embedded in a hydrophobic environment[41], such as in cloud droplets, atmospheric ice particles[1], rocks[3] or medicine carriers[9] is lower than what one might think based on the actual temperature. Future studies on such nanoscopic systems may reveal important information on the role and reactivity of interfacial water in aerosol formation, protein folding, pore functioning[4,6,7,42] and the charging and stabilization of hydrophobic interfaces.

## Methods

**Chemicals.** Before use all glassware was cleaned with a 1:3 $H_2O_2$:$H_2SO_4$ solution, after which it was thoroughly rinsed with ultra-pure water ($H_2O$, Milli-Q UF plus, Millipore, Inc., electrical resistance of 18.2 MΩ cm). Hexadecane (Fluka, 99.8%), decane (Fluka, 99.8%), cyclohexane (Sigma, >99.7%), Span80 (Sigma, GC quality), Tween80 (Sigma, GC quality), sodium dodecyl sulfate, SDS (99% BioMol), $d_{34}$-hexadecane ($C_{16}D_{34}$, 98% d, Cambridge Isotope Laboratories), fluorinated oil (Novec HFE7500, 3-Ethoxy-dodecafluor-2-trifluormethyl-hexan), sulfuric acid (95–97%, ISO, Merck) and $H_2O_2$ (30%, Reactolab SA) were used as received. The purity of alkanes was verified with a Zisman test[43,44]. All samples for SFS measurements were prepared using $D_2O$ (99.8% Armar, >2 MΩ cm). The $D_2O$ for the water/air SFG experiment was obtained from Cambridge Isotope Laboratories (99.9%).

**Water nanodroplets.** Water nanodroplets were prepared using a sonication procedure with 1 v.v.% $D_2O$ in oil concentration and 5 mM Span80 (or Span80:Tween80 mixture), according to the procedure described in ref. 45. For some samples (shown in Supplementary Fig. 2) 10 mM SDS was added to the water phase before the emulsification. The droplet-size distribution was measured with dynamic light scattering (DLS, Zetasizer Nano ZS, Malvern). For all samples, the droplets have a mean diameter of ~200 nm with a polydispersity index of ~0.2. Infrared spectra were recorded with a Bruker Vertex 70 instrument equipped with silver mirrors.

**Monocrystalline ice.** Monocrystalline ice was grown following the manual of Basu et al.[46]. Five pieces of PVC pipe with small notches at the bottom were placed in an aluminium pan, which was chilled from underneath with a cooling liquid. Millipore water (18.2 MΩ cm) was degassed and added until the bottoms of the moulds were covered. The temperature in the pan was set to −0.5 °C and after equilibration ice nucleation was initiated by adding a small piece of ice to the centre of the pan. The small notches at the bottom of each PVC mould allow only one ice crystal to propagate through, resulting in a single ice crystal in each mould. For 3 subsequent days, 1 cm water was added to each mould and the temperature was lowered by 0.3 °C. At day 4, the ice was pushed outside of the pipe pieces and stored in a freezer, where all further handling was done. The crystal orientation was determined using a Rigsby-type universal stage[47] and cut to the basal plane with a band saw. The ice surface was polished by a microtome and a 4 mm thick slice was cut off for the experiment.

**Vibrational sum frequency reflection mode spectra.** The SFG and VIS beams were polarized in the vertical (S) direction, and the infrared beam was polarized in the horizontal plane (P), leading to the SSP (SFG, VIS, IR) polarization combination using the experimental set-up described in ref. 26. The set-up was purged with $N_2$ gas. The VIS beam with a pulse energy of 14 μJ was centred at 798.6 nm with a full-width at half-maximum (FWHM) bandwidth of 13 $cm^{-1}$. The $D_2O$/air measurements were performed with the 10 μJ 300 $cm^{-1}$ FWHM infrared beam centred at a wavelength of 4 μm. The procedure for measuring hexane/water is given in ref. 26. To measure the water/Span80 interface, 0.16 μM Span80 was dissolved in 0.1 ml hexane and added to liquid $D_2O$. Subsequently, the hexane was

evaporated off. The diameter of the teflon tray for this experiment was 3.5 cm. The basal ice/air measurements were performed with a 3 µJ 550 cm$^{-1}$ FWHM IR beam centred at 3 µm. The angles of incidence with respect to the surface normal were 35° (VIS) and 40° (infrared). The D$_2$O/air and basal ice spectra were recorded with acquisition times of 30 s and 300 s, respectively. The spectral intensities were baseline subtracted and normalized with the infrared and VIS pulse energies, and the (normalized) SFG spectrum recorded from z-cut quartz.

The ice measurements were performed using a temperature cell that was cooled with liquid N$_2$. It allows transmittance of the laser beams through a CaF$_2$ window (see Supplementary Fig. 4). The temperature was monitored by a thermocouple welded on the edge of the ice surface with a drop of water and the desired temperature was set by a heating foil resistance, covered by a copper plate.

**Vibrational SFS spectra.** Vibrational SFS spectra were recorded using the set-up for SFG experiments described in refs 48–50. An 800 nm regeneratively amplified Ti:sapphire system (Spitfire Pro, Spectra physics) seeded with an 80 MHz 800 nm oscillator (Integral 50, Femtolasers) was operated at a 1 kHz repetition rate to pump a commercial OPG/OPA/DPG system (HE-TOPAS-C, Light Conversion), which was used to generate infrared pulses. The visible beam was split off directly from the amplifier, and spectrally shaped with a home-built pulse shaper. The angle between the 10 µJ visible (VIS) beam (800 nm, FWHM 15 cm$^{-1}$) and the 6 µJ infrared beam (3–4.5 µm, FWHM 160 cm$^{-1}$) was 20° (as measured in air). The focused laser beams were overlapped in a sample cuvette with a path length of 200 µm. At a scattering angle ($\theta$, measured in air) of 55°, the scattered sum frequency (SF) light was collimated using a plano-convex lens ($f = 15$ mm, Thorlabs LA1540-B) and passed through two short-wave pass filters (3rd Millenium, 3RD770SP). The SF light was spectrally dispersed with a monochromator (Acton, SpectraPro 2300i) and detected with an intensified CCD camera (Princeton Instruments, PI-Max3) using a gate width of 10 ns. The acquisition time for a single spectrum was 300 s. A Glan-Taylor prism (Thorlabs, GT15-B), a half-wave plate (EKSMA, 460-4215) and a polarizing beam splitter cube (CVI, PBS-800-050) and two BaF$_2$ wire grid polarizers (Thorlabs, WP25H-B) were used to control the polarization of the SFG, VIS and infrared beams, respectively. All measurements were performed in the SSP polarization combination. For normalization purposes, a reflection beam path is accessible by removable optics so that scattering and reflection can be done without a realignment procedure.

**Temperature-controlled SFS measurements.** Temperature-controlled SFS measurements were performed using a custom-made sample cell (Quantum Northwest, Supplementary Fig. 5). The sample cuvette was placed in a metallic holder cooled with a single Peltier element. The sample chamber was closed and filled with N$_2$ gas to avoid condensation of air. The windows of the temperature cell were made from CaF$_2$ (for the incoming beams) and quartz (for the scattered SF beam). The temperature cell itself was thermally isolated and also placed inside a box purged with N$_2$ gas. The hot side of the Peltier element was cooled with a flow of cold liquid ethanol (223 K, using a peristaltic pump, Ismatec ISM1200). The cold ethanol was obtained by cooling ethanol inside a copper coil that was placed in a bath containing a mixture of dry ice and ethanol. The infrared beam path outside the laser source was purged with N$_2$ gas.

**Generation of broadband infrared pulses.** To probe the whole frequency range of the D$_2$O stretch mode region for the SFS spectral recordings, we stepped the infrared frequency trough the spectral window between 2,000 and 3,400 cm$^{-1}$ using 50 nm steps (leading to a total acquisition time of 5,400 s per spectrum). Supplementary Fig. 6a shows SF spectra for the different infrared centre frequencies recorded in reflection mode from a z-cut quartz crystal. The summed spectrum for one of the runs is shown in Supplementary Fig. 6b.

**Spectral shape and infrared absorption.** The scattered SF spectra recorded from the water droplets were baseline subtracted and summed to obtain the spectral intensity for the entire spectral region of the OD stretching mode. The resulting spectrum was then divided by the energies of the infrared and VIS beams and normalized to the summed SFG spectrum of a z-cut quartz crystal (shown for example in Supplementary Fig. 6b). Since SF scattering experiments are performed in transmission mode, we have to take into account that the oil phase is absorbing part of the infrared spectrum, leading to a transmission spectrum $\mu_{FTIR}$. To correct for absorption effects, we divide the obtained SFS spectrum by a Fourier transform infrared transmittance spectrum measured from the same sample in the same cuvette. Supplementary Fig. 7 shows the results of this correction as the blue spectra. It can be seen that this procedure results in significant spectral changes in the red spectral side since here the SFG spectrum overlaps with the CD modes of the deuterated main phase (d$_{34}$-hexadecane). The CH modes in the SFS spectrum originate from the Span80 molecules that are present on the oil side of the interface (Span80 is insoluble in water[18]).

**Discontinuity in the interfacial electric fields.** Frequency-dependent refraction modifies the incident electromagnetic field at the interface, which can lead to significant frequency-dependent distortions in the obtained water spectra[51,52]. The

sum frequency reflection mode spectra are therefore divided by the Fresnel factors[14], whereby we use the refractive indices of D$_2$O and ice from refs 53–55. The effective refractive index of the interfacial layers is calculated using a simple slab model[56]. For linear light scattering experiments such effects are captured by linear Mie theory[57] and thus for SFS experiments by nonlinear Mie theory[58]. Using nonlinear Mie theory, we calculated a frequency-dependent correction factor for the SFS spectra using the input parameters from Supplementary Table 2. The refractive index values for oil and D$_2$O are taken from refs 59,60. Supplementary Fig. 8 shows the effects of the correction on the spectral data reported in Figs 1 and 2, where it can be seen that the influence of refraction effects is minimal. The reason for the small difference is that the droplets are small compared to the wavelength. For simplicity, in the main text we refer to the corrected SFS intensity $I_{SFS}/I_{IR}\mu_{FTIR}$ as normalized SFS intensity.

**Data availability.** The data that support the findings of this study are available from the corresponding author on reasonable request.

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

## Acknowledgements

This work is supported by the Julia Jacobi Foundation, and the European Research Council (Grant number 616305). We thank Prof. Esther Amstad for providing the sample of water droplets in fluorinated oil. Dion Ursem and Hinco Schoenmaker are acknowledged for technical support. Part of this work belongs to the research programme of the Netherlands Organisation for Scientific Research (NWO) and was performed at the research institute AMOLF.

## Author contributions

N.S. and W.J.S. performed experiments, N.S., W.J.S., H.J.B. and S.R. interpreted the data and wrote the manuscript. S.R. conceived and supervised the work.

## Additional information

**Competing interests:** The authors declare no competing financial interests.

