## [Peer Review File · Nature Communications]

Editorial Note: Parts of this peer review file have been redacted as we could not obtain permission to publish the reports of reviewer 2.

Reviewers' comments:

Reviewer #1 (Remarks to the Author):

This is an interesting article that involves some very experimentally difficult studies. The Roke group has pioneered this technique which is a significant achievement.

That said, there are some serious issues of concern that need to be addressed before this can be considered for publication in Nature.

1. The title is misleading. It should be something closer to "The interfacial structure of surfactant stabilized water droplets in a hydrophobic liquid". At first glance at the title I thought I would be reviewing a paper on water droplets suspended in air. Such was not the case.

2. The spectra for the droplet and the air/water interface are provided. However, there are problems with the interpretation. Based on years of extensive previous studies by the Richmond group on planar water/hydrophobic oils surfaces, any trace amount of surfactant (ionic or nonionic) greatly alters the interfacial water structure at the hydrophobic oil/water interface. The dangling OH (or OD) bond in this case disappears in the presence of small amounts of surfactant, especially a charged one, and the spectral signatures for the more strongly hydrogen bonded modes are greatly enhanced to the point that the interfacial water structure looks more like ice for the water/hydrophobic liquid interface. Even nanomolar concentrations of impurities cause this to happen – resulting in a spectrum that looks similar to what is shown in this paper for the nanoemulsions. The authors indeed state that the "The absence of unbonded OD groups in the droplet spectrum can be explained by the presence of OH groups on the Span80 molecule that can form H-bonds with interfacial water molecules that would otherwise be unbonded. As a result, the peak at 2745 cm⁻¹, corresponding to OD groups that re not H-bonded vanishes." So they admit that the Span80 has altered the surface of water for the unbonded OD but then ignore that it would also alter the hydrogen bonded region even though they admit that Span80 has OH groups that DO strongly bond to hydrogen bonded water. The SI tries to show that it has no effect for the presence of SPAN80 or SDS but this is not convincing. An interesting experiment to test whether SDS is there would be to use deuterated SDS where the CD stretches would show up as opposed to the current experiment where they are in a different spectral region.

What is missing throughout the paper is a discussion of the interface as one that is comprised of water, Span80, charged surfactant SDS and a hydrophobic oil – not merely a hydrophobic oil and water, and how this more complex interface is affected by temperature including how the Span80 might restructure with temperature and SDS in cases where it is present. If a comparison is made to the D₂O/air interface then it should be D₂O/air with a Span80 and SDS added at comparable concentrations.

Or alternatively to do the study of just water and the hydrophobic oil – which is likely challenging to get it to stabilize.

3. With the following claim "From the detection limit of the SFS system it follows that there are fewer than 1 free OD groups per 27nm² pointing out of the water." It should be again reiterated that this is due to the surfactants there and is not necessarily indicative of a true hydrophobic oil/water interface.

4. If the claim is made that even with the Span80 and SDS at the interface that this still represents a true hydrophobic oil/water interface then the authors need to explain in more detail why the spectra are so much different from the large number of studies of oil/water planar interfaces by the Richmond group that show that any type of surfactant at the oil/water interface significantly perturbs surface water molecules, particularly trace amounts of charged surfactants which produce effects of a loss in unbonded OD and an enhancement in the spectral signatures for the bonded water molecules.

Once these issues are addressed then the decision needs to be made as to whether this work is novel enough to warrant publication in Nature. This is a very interesting piece of work and is worth publishing as long as the arguments for the interpretation are sound.

If it was measurements of what I initially thought the article was about based on the title, it would certainly be the case that it was appropriate for Nature.

We would like to thank all reviewers for carefully reading our manuscript and for providing helpful comments and suggestions. We have revised the manuscript on the basis of the comments and provide detailed replies below. The original comments of the referees are in *italic* and our responses are in normal font.

Reviewer #1 (Remarks to the Author):

This is an interesting article that involves some very experimentally difficult studies. The Roke group has pioneered this technique which is a significant achievement.

That said, there are some serious issues of concern that need to be addressed before this can be considered for publication in Nature.

1. The title is misleading. It should be something closer to “The interfacial structure of surfactant stabilized water droplets in a hydrophobic liquid”. At first glance at the title I thought I would be reviewing a paper on water droplets suspended in air. Such was not the case.

Following the suggestion of the reviewer we have changed the title to: “The interfacial structure of water droplets in a hydrophobic liquid”.

2. The spectra for the droplet and the air/water interface are provided. However, there are problems with the interpretation. Based on years of extensive previous studies by the Richmond group on planar water/hydrophobic oils surfaces, any trace amount of surfactant (ionic or nonionic) greatly alters the interfacial water structure at the hydrophobic oil/water interface. The dangling OH (or OD) bond, in this case, disappears in the presence of small amounts of surfactant, especially a charged one, and the spectral signatures for the more strongly hydrogen bonded modes are greatly enhanced to the point that the interfacial water structure looks more like ice for the water/hydrophobic liquid interface. Even nanomolar concentrations of impurities cause this to happen – resulting in a spectrum that looks similar to what is shown in this paper for the nanoemulsions. The authors indeed state that the “The absence of unbonded OD groups in the droplet spectrum can be explained by the presence of OH groups on the Span80 molecule that can form H-bonds with interfacial water molecules that would otherwise be unbonded. As a result, the peak at 2745 cm⁻¹, corresponding to OD groups that re not H-bonded vanishes.”

So they admit that the Span80 has altered the surface of water for the unbonded OD but then ignore that it would also alter the hydrogen-bonded region even though they admit that Span80 has OH groups that DO strongly bond to hydrogen bonded water. The SI tries to show that it has no effect for the presence of SPAN80 or SDS but this is not convincing. An interesting experiment to test whether SDS is there would be to use deuterated SDS where the CD stretches would show up as opposed to the current experiment where they are in a different spectral region.

What is missing throughout the paper is a discussion of the interface as one that is comprised of water, Span80, charged surfactant SDS and a hydrophobic oil – not merely a hydrophobic oil and water, and how this more complex interface is affected by temperature including how the Span80 might restructure with temperature and SDS in cases where it is present.

If a comparison is made to the D2O/air interface then it should be D2O/air with a Span80 and SDS added at comparable concentrations.

Or alternatively to do the study of just water and the hydrophobic oil – which is likely challenging to get it to stabilize.

This remark contains the following questions / concerns of the reviewer:

1. What is the influence of impurities on the experiment as they are known to be important for past SFG reflection experiments?
2. There is a concern that we are not comparing the right interfaces; oil and air are not comparable and surfactant (Span80) and oil are different.
3. The referee asks for a discussion of the SDS / Span80 / hexadecane / water mixture and the effect of SDS in general.
4. The referee asks whether Span80 restructures with temperature.

Ad 1.

Impurities are indeed a major concern for interface studies. There is, however, a significant difference between reflection mode and scattering experiments. Given a certain amount of impurity, the influence it has on the experiment is determined by (a) the volume of chemicals used, (b) the preparation procedure, (c) the available surface to volume ratio, and (d) whether impurities partition to the interface.

(a) - The volume of our samples (and used chemicals) is low compared to planar interface experiments. Scattering experiments employ 60 μL of liquid, reflection mode experiments require several mLs at least, so the total amount of impurities is significantly less for SF scattering experiment.

(b) – For SFS experiments all chemicals are mixed in the same vial, and are not exposed to air.

(c) – The available surface to volume ratio is ~ 1000 times bigger for SF scattering than for SF reflection experiments. See Ref. [1] for more details. SFS is thus far less affected by potential impurities than conventional reflection SFG. This is one of the strongest advantages of SFS over conventional SFG.

(d) – The partition of impurities to the interface does not need to be the same for planar and droplet experiments. Previously we have shown this to be the case [2]. Furthermore, water droplets cannot be stabilized by SDS [3, 4].

In addition chemicals of highest purity were used and tests were made to verify that the purity is indeed maximum (e.g. a Zisman test, see materials and methods).

Ad. 2.

We have addressed this point by including in the manuscript SFG spectra of hexane / D_2O (Figure 1), and Span80 / D_2O at a saturated concentration (Figure S8). For comparison, Fig. R1 shows all 4 spectra together. We have added more explanatory text to page 3 and 4, when comparing the reflection mode and scattering mode spectra.

As Figure R1 shows the reflection mode Span80/water and hexane/water spectra both have a free OD vibrational peak, which is greatly reduced in intensity compared to the water/air interface, but it is still present. Furthermore, the Span80/ D_2O and hexane/ D_2O spectra have a similar shape compared to the air/ D_2O interface spectrum in the OD-vibrational region. The water/hexane interface SFG spectrum has a somewhat enhanced strongly H-bonded peak compared to that on air and Span80/ D_2O , but this enhancement is much smaller than what we observe in the SFS spectra of water droplets dispersed in a hydrophobic liquid mixture of hexadecane and Span80. Therefore, the presence of oil or Span80 cannot give rise to the strong enhancement in interfacial H-bonding that is observed in the case of droplets.

In addition, we note in relation to the concern about the effect of charged surfactants on the water spectrum, that the air/SDS/water spectrum published in [5], Figure 2, does not display an enhancement of H-bonding but rather a weakening.

Figure R1. Comparison of SFS spectrum of water droplets in a hydrophobic liquid (black line, 1vol% D₂O in 5mM Span80 in d₃₄-hexadecane) with SFG reflection spectra from D₂O/air (blue), D₂O/hexane (green) and D₂O/Span80 (red) interface.

Ad. 3

Water droplets cannot be stabilized by SDS [3, 4] and there is no indication that there is any specific SDS impurity in the used chemicals – it is hard to see where it would come from. As shown in Fig. S7 and now also in Fig. 2 various oils and surfactants all lead to the same features in the SFS droplet spectra of water, meaning that the observed enhancement of the hydrogen-bond structure of the water droplet surface is not related to the interaction with the surfactant, but to the interaction with the aliphatic chains. As also noted by the referee it is impossible to prepare water droplets in alkanes, one has to resort to a mixture of hydrophobic liquids that do allow the production of water droplets.

Nevertheless, we have addressed the concern about the relevance of SDS in more detail by measuring the SFS spectrum of the D₂O droplets prepared in h-hexadecane / Span80 with 10 mM d₂₅-SDS added to the water phase. The data is shown in Fig. R2. It can be seen that neither the SO stretch region nor the CD stretch region displays any features of DS⁻ anions. This agrees with stability data from the late 1940's [3, 4] that reported on the incapability of hydrophilic (or charged) surfactants to stabilize water droplets in oil. Indeed if this would be the case, there would be a surface of negative charge on the water surface, which would lead to a very strong Coulombic repulsion at the interface and also within the droplet (as the counter ions would have to be in the bulk). Please also note that in the reversed system, hexadecane droplets of ~100 nm in D₂O, tiny amounts of DS⁻ ions can be detected, even at 10 μM concentrations of SDS [2].

This confirms that SDS is a surfactant stabilizing oil droplets in water and not water droplets in oil [3, 4]. The result of Fig. R2 suggests that other similar anionic impurities will likely also not be

interfacially active in the water droplet / oil systems studied here. A change in the surface structure of water induced by charged surfactants is therefore highly unlikely.

[Redacted]

Ad. 4

According to the reviewer's suggestion, we have measured the SFS spectra of water droplets focusing on the CH modes of Span80 at different temperatures (corresponding to the water spectra shown in the paper). As can be seen from Figure R3 we do not observe significant changes in the interfacial Span80 structure.

Figure R3. SFS spectra of D₂O droplets in d₂₂-decane with 5mM Span80 in the CH-vibrational region at room temperature (296 K, red) and under supercooled conditions (green, 273 K and blue, 263 K).

3. With the following claim “From the detection limit of the SFS system it follows that there are fewer than 1 free OD groups per 27nm² pointing out of the water.” It should be again reiterated that this is due to the surfactants there and is not necessarily indicative of a true hydrophobic oil/water interface.

The absence of a free OD peak cannot be entirely due to the presence of the surfactant because the water/Span80 spectrum in fact does have a free OD peak. We rather relate the absence of a free OD peak to the structural differences between a curved water droplet embedded in a hydrophobic liquid and a planar oil/water interface. We have added text on page 4 to clarify this point.

4. If the claim is made that even with the Span80 and SDS at the interface that this still represents a true hydrophobic oil/water interface then the authors need to explain in more detail why the spectra are so much different from the large number of studies of oil/water planar interfaces by the Richmond group that show that any type of surfactant at the oil/water interface significantly perturbs surface water molecules, particularly trace amounts of charged surfactants which produce effects of a loss in unbounded OD and an enhancement in the spectral signatures for the bonded water molecules.

We hope our answer to question 2 has addressed this concern. We now make a comparison with all three relevant systems air/water, hexane / water and Span80 / water. We also note that the planar oil/water system was re-measured with fs SFG recently and that differences with the Richmond group spectra were found. We refer to Ref. [6] for a discussion. As for the current work, we have supplied all measured data for comparison.

Once these issues are addressed then the decision needs to be made as to whether this work is novel enough to warrant publication in Nature. This is a very interesting piece of work and is worth publishing as long as the arguments for the interpretation are sound.

If it was measurements of what I initially thought the article was about based on the title, it would certainly be the case that it was appropriate for Nature.

We hope our answers have addressed all the concerns of the referee and we think they have helped much to improve the quality of the manuscript.

References

1. Samson, J.-S., et al., *Sum frequency spectroscopy of the hydrophobic nanodroplet/water interface: Absence of hydroxyl ion and dangling OH bond signatures*. Chemical Physics Letters, 2014. **615**: p. 124-131.
2. Aguiar, H.B.d., et al., *The Interfacial Tension of Nanoscopic Oil Droplets in Water Is Hardly Affected by SDS Surfactant*. Journal of the American Chemical Society, 2010. **132**(7): p. 2122-2123.
3. Ho, O.B., *Electrokinetic Studies on Emulsions Stabilized by Ionic Surfactants: The Electroacoustophoretic Behavior and Estimation of Davies' HLB Increments*. Journal of Colloid and Interface Science, 1998. **198**(2): p. 249-260.

4. Griffin, W.C., *Classification of surface-active agents by "HLB"*. Journal of Cosmetic Science, 1949. **1**(5): p. 311-326.
5. Livingstone, R.A., et al., *Two Types of Water at the Water–Surfactant Interface Revealed by Time-Resolved Vibrational Spectroscopy*. Journal of the American Chemical Society, 2015. **137**(47): p. 14912-14919.
6. Strazdaite, S., et al., *Enhanced ordering of water at hydrophobic surfaces*. The Journal of Chemical Physics, 2014. **140**(5): p. 054711.

Reviewers' comments:

Reviewer #1 (Remarks to the Author):

This is an improved paper but the rebuttal to a number of the reviewer concerns are still not convincing and the rewrite continues to include overselling of the observations and results with the intention of justifying its publication in Nature.

2 of the more serious examples.

In the introduction: "nanoscopic water droplets in a liquid hydrophobic environment, which can be considered as a *realistic model* for an aerosol".

Atmospheric aerosols and marine aerosols are packed with inorganic salts and strong acids that can significantly alter the behavior of the oil/water surface and surfactants used in these studies. And coated with small organic compounds, not covered with long chain surfactants and polymers such as in these studies. So to assume that these studies are a realistic model or are representative of marine aerosols is a strong exaggeration and will not be acceptable to many who study atmospheric aerosol particles.

In the conclusions: The presented experiments... illustrate the *large impact* of nanoscopic hydrophobic surfaces on the H-bond structure of water". The conclusions are drawn largely from a very simplistic analysis of the increase in the OH stretch modes at lower frequencies. There is nothing quantitative in the measurements to justify that this increase illustrates a "large" impact on the surface. Especially given the nonlinearity in the optical response. Just because there is an observable increase in the intensity is not necessarily indicative of a "large impact".

The authors have worked to justify the issues of impurities but still ignore some of the literature involving impurities in SDS as an example. But the impurity issue is not the main reason for rejecting the paper, the more worrisome issues are those above.

We thank the referees for examining our manuscript in detail. Referee comments are in *blue*, the answers in black.

Reviewer #1 (Remarks to the Author):

This is an improved paper but the rebuttal to a number of the reviewer concerns are still not convincing and the rewrite continues to include overselling of the observations and results with the intention of justifying its publication in Nature.

2 of the more serious examples.

*In the introduction: "nanoscopic water droplets in a liquid hydrophobic environment, which can be considered as a *realistic model* for an aerosol".*

Atmospheric aerosols and marine aerosols are packed with inorganic salts and strong acids that can significantly alter the behavior of the oil/water surface and surfactants used in these studies. And coated with small organic compounds, not covered with long chain surfactants and polymers such as in these studies. So to assume that these studies are a realistic model or are representative of marine aerosols is a strong exaggeration and will not be acceptable to many who study atmospheric aerosol particles.

Following the suggestion of the reviewer we have removed the reference to aerosols in the introduction and in the conclusion.

*In the conclusions: The presented experiments... illustrate the *large impact* of nanoscopic hydrophobic surfaces on the H-bond structure of water". The conclusions are drawn largely from a very simplistic analysis of the increase in the OH stretch modes at lower frequencies. There is nothing quantitative in the measurements to justify that this increase illustrates a "large" impact on the surface. Especially given the nonlinearity in the optical response. Just because there is an observable increase in the intensity is not necessarily indicative of a "large impact".*

We quantify the changes in the SFG spectrum of a water droplet by comparing the measured spectrum with SFG spectra of a planar water interface recorded at various temperatures. A droplet surface has the same molecular ordering as a planar surface that is 50 K colder.

The nonlinear nature of SFG is not of relevance for this comparison as we compare SFG data with SFG data. Indeed, if we would have compared SFG spectra from a droplet surface to an IR or Raman spectrum from bulk water, this would have been an issue. Whether 50 K is a lot or not, is of course a matter of taste. To formulate the conclusions in a more objective manner, we removed the words "large impact" from the text.

The authors have worked to justify the issues of impurities but still ignore some of the literature involving impurities in SDS as an example. But the impurity issue is not the main reason for rejecting the paper, the more worrisome issues are those above.